# Social-media based Health Education plus Exercise Programme (SHEEP) to improve muscle function among community-dwelling young-old adults with possible sarcopenia in China: A study protocol for intervention development

Ya Shi [1,2]*, Emma Stanmore[1,3,4,5], Lisa McGarrigle[1,3,4], Chris Todd[1,3,4,5]

1 School of Health Sciences, Faculty of Biology, Medicine & Health, The University of Manchester, Manchester, United Kingdom, 2 School of Nursing & School of Public Health, Yangzhou University, Yangzhou, Jiangsu Province, China, 3 Manchester Institute for Collaborative Research on Ageing, Manchester, United Kingdom, 4 Manchester Academic Health Science Centre, Manchester, United Kingdom, 5 Manchester University NHS Foundation Trust, Manchester, United Kingdom

* ya.shi@postgrad.manchester.ac.uk

## Abstract

Possible sarcopenia refers to low muscle strength. Prevalence of possible sarcopenia is estimated to be significantly higher in community-dwelling older adults than that of confirmed or severe sarcopenia. However, there are currently far fewer non-pharmacological intervention strategies for possible sarcopenia than for sarcopenia in the community. Meanwhile, one type of non-pharmacological intervention in sarcopenic area, health education, is under-researched, and older people's awareness about sarcopenia is extremely low, necessitating an immediate dissemination tool for prevention. Social media may be a potential, scalable, low-cost tool for this. This study protocol outlines how a social media-based multicomponent intervention will be co-designed with stakeholders to address this evidence gap. Guided by the Medical Research Council's framework, the proposed research covers two phases that employ a co-design approach to develop a theory-based multicomponent intervention to increase sarcopenia prevention in the community. The participants will be recruited from young-old adults (60~69) with possible sarcopenia in the community of Changsha, China. Maximum sample size will be 45 participants in total, with 18~25 participants in the development phase and 15~20 participants in the pre-test phase. During two rounds of focus groups with older adults, a social-media based intervention strategy will be developed from a theory-based conceptual model and an initial intervention plan formulated by the research group. After this, there will be a three-week pre-test phase, followed by a semi-structured interview to further modify the theory-based conceptual model and the social-media based intervention strategy. The focus of the data analysis will be on thematic analysis of qualitative data primarily derived from the group interview and the semi-structured interview with key stakeholders.

**Data Availability Statement:** Deidentified research data will be made publicly available when the study is completed and published.

**Funding:** Y.S. is funded by the University of Manchester - China Scholarship Council Joint Scholarship (202108320049), and C.T. is funded by the National Institute for Health and Care Research Senior Investigator Award (NIHR200299). The funders did not have any role in study design, data collection and analysis, decision to publish, or preparation of the manuscript.

**Competing interests:** The authors have declared that no competing interests exist.

## Introduction

Possible or probable sarcopenia refers to low muscle strength, which is a relatively new classification proposed by the European Working Group on Sarcopenia in Older People (EWGSOP2, 2018) [1] and the Asian Working Group for Sarcopenia (AWGS, 2019) [2]. The prevalence of possible sarcopenia is quite high among community-dwelling older adults. According to a latest population-based longitudinal study conducted in China, the estimated prevalence of older adults aged 60 years and over with possible sarcopenia in the community was 46.0% [3]. Pérez-Sousa et al. [4] investigated 5237 Colombian older adults aged≥60 years and found that the prevalence of probable sarcopenia was as high as 46.5%. Several recent cross-sectional studies also reported a high prevalence of possible sarcopenia in older population living in the community, with 26.9% in Swiss (>60y) [5], 25.4% in Greek (≥75y) [6], 23.7% in Korean (≥65y) [7]. Besides, Wu et al. [8] discovered that the prevalence of possible sarcopenia (38.5%) among Chinese older people in the community was significantly higher than that of confirmed (18.6%) and severe (8.0%) sarcopenia.

The young-old age group has a high prevalence of possible sarcopenia, but relatively fewer intervention methods, requiring additional focus. Up to 71% of the participants in the longitudinal study conducted in China who reported possible sarcopenia were between 60 and 70 years of age [3]. More than half (52%) of the older Colombians diagnosed with possible sarcopenia in the cross-sectional study were aged 60 to 69 years [4]. Wu et al. [9] examined that the prevalence of possible sarcopenia among adults aged 55 years and older at 11 rural community daycare centres in Taiwan was as high as 68.7%. In addition, our recent scoping review about the non-pharmacological interventions for possible sarcopenia or sarcopenia revealed that 70-79-year-olds (64.8%) have been studied more than 60-69-year-olds (18.5%) in community-dwelling older adults, and the number of interventions for possible sarcopenia (11.9%) was significantly lower than for sarcopenia (72.9%) [10]. However, both the EWGSOP2 and AWGS2019 recommend possible sarcopenia as an important threshold to trigger assessment of causes and initiate intervention in medical practice [1,2].

Non-pharmacological interventions are essencial for sarcopenia prevention, but health education is under-researched and public awareness about sarcopenia is extremely low. Health Education, as defined by the World Health Organization, refers to the deliberate creation of learning opportunities that involve communication aimed at enhancing health literacy. This includes improving knowledge and developing life skills that promote individual and community health [11]. A cross-sectional study examined the current knowledge of middle-aged and older adults in Netherlands regarding sarcopenia and showed that only 9% of participants (57.0~75.1y) reported knowing what sarcopenia is [12]. EWGSOP2, AWGS2019, and some researchers recommend facilitating timely lifestyle interventions and related health education for primary health care in community and prevention settings, which will increase awareness of sarcopenia prevention and intervention in diverse health care settings [1,2,13–15]. Nevertheless, our scoping review found that there were 142 intervention groups designed in 59 studies for preventing possible sarcopenia or sarcopenia in the community, but the proportion of intervention groups containing health education components (15.5%) was substantially lower than those containing exercise or nutrition components (52.8% and 34.5%, respectively) [10]. In 22 intervention groups containing health education components, up to 18 groups (81.8%) provided education materials unrelated to sarcopenia, which may be the primary reason for unsatisfactory results of existing health education [10]. Furthermore, this scoping review identified only three traditional forms of health education, including group-based classes, face-to-face interactions and leaflets/materials [10], which appears to be significantly fewer than the forms of health education in other chronic diseases, like digital

health education on diabetes management [16] and social media as an educational platform on hypertension [17–19]. An integrative review indicated that technologies promoting health education for older adults in the community also included software, videos, mock-up and so on [20].

In the context of an Internet era and in light of the increased use of digital tools during the covid-19 pandemic, social media may be a promising medium for health education to disseminate knowledge and raise awareness of disease prevention among older adults. For instance, a pilot study in United States demonstrated that a social media-based (WeChat) health education was feasible, acceptable, and potentially efficacious in self-management education of older adults with type 2 diabetes [21]. Another qualitative research found that a photography-based, social media (Facebook) walking game was well received by older women and also benefited for increasing their motivation for physical activity, fostering a shift in perspective, increasing their knowledge, and providing health-related benefits to them [22]. TikTok, a platform for sharing short videos, has been phenomenal since the onset of the COVID-19 pandemic and has reached Chinese seniors [23] by providing simple video-editing tools [24] and catering to specific informational and practical needs [25,26]. A qualitative research showed that during the COVID-19 pandemic, WeChat and TikTok played a significant role in providing older Chinese adults with access to valuable health information; and many older adults were able to modify their health behaviours after incorporating this information and knowledge into their daily lives [27]. Kassamali et al. [28] reported that TikTok held great potential as a platform for disseminating educational information about disease, as their study indicated that the number of views for the hashtag such as #acne, #alopecia, #cyst, #rosacea, and #psoriasis on TikTok doubled in just 5 months. Nonetheless, there are currently no studies utilizing social media for sarcopenia prevention. Hence, TikTok may be a valuable platform for attempting to develop a short-video-based intervention strategy to prevent sarcopenia in Chinese older adults.

This research programme was founded on behaviour change theories. Previous researches have already demonstrated that exercise could improve sarcopenic indices in sarcopenia patients [29–31], but the maintenance of improvements in long-term behaviour change in community-dwelling older adults with sarcopenia is unclear [10]. Strong evidence indicates that successful behaviour change interventions in physical activity include several key components, such as theoretical framework, useful intervention components, and behaviour change techniques [32–35]. Therefore, this study constructed a new conceptual model primarily guided by Michie et al.'s Behaviour Change Wheel (BCW) [32] and the conceptual model for Lifestyle- integrated Functional Exercise using smartphones and smartwatches (eLiFE) [33], with the objective of developing intervention content and monitoring process. BCW is a framework (S1 Fig) [32] mainly for developing intervention content for behaviour change, and several studies have demonstrated its efficacy and recommended its use in interventions for older adults [36,37]. In addition, the intervention phase was derived from the eLiFE conceptual model (S2 Fig) [33], which assists older adults in forming long-term physical activity habits through the use of a mobile app, and the eLiFE programme was shown to be feasible and safe for young seniors [38]. We integrated the BCW and eLiFE conceptual models to create a new SHEEP conceptual model for preventing sarcopenia among older people (Fig 1).

To improve the acceptability of the theory-based intervention strategy among older individuals, co-design method will be employed. Strong evidence found that patients (even those close to death) and their families were consistently willing to engage in research [39]. Co-design in health care services typically refers to active collaboration between researchers, specialists, health care professionals, and non-academic partners such as patients or family members who are regarded as 'experts of their experiences' [40–42]. In recent years, co-design

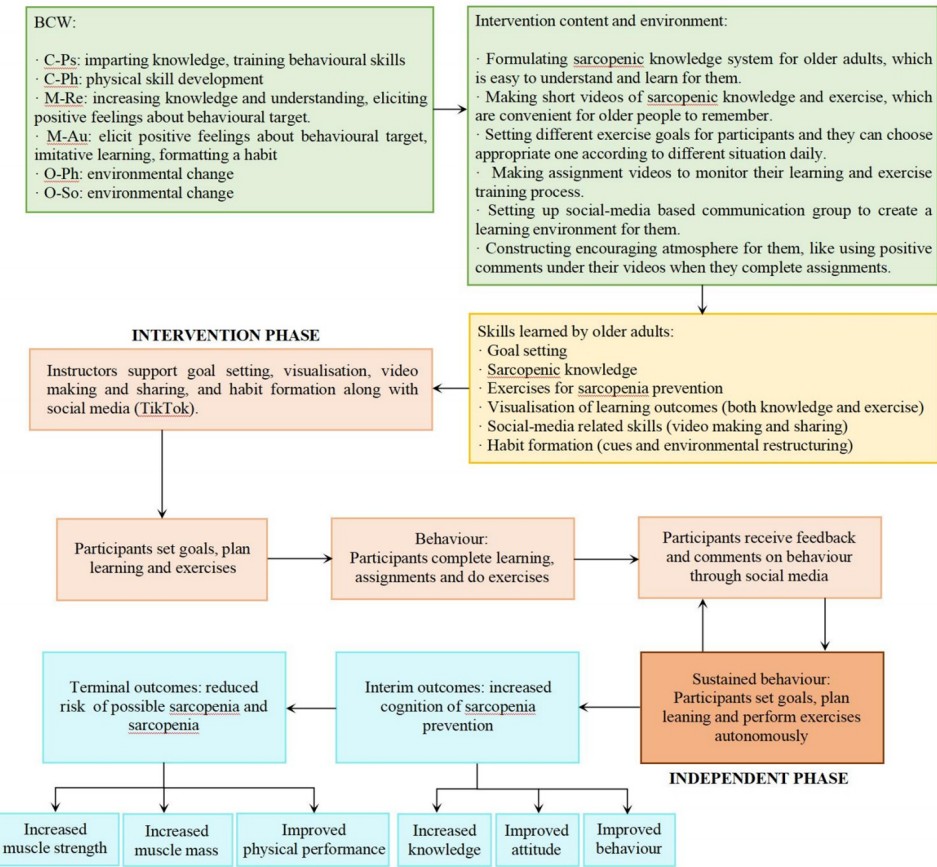

**Fig 1. SHEEP conceptual model.** This is the initial version of the model, which was plan to undergo further modification as the research advances. The complete designations of the abbreviations in this model are as follows: C-Ps, psychological capability; C-Ph, physical capability; M-Re, reflective motivation; M-Au, automatic motivation; O-Ph, physical opportunity; O-So, social opportunity.

methods have been utilised extensively in health care interventions, for example, a co-designed mHealth programme to promote healthy lifestyles [43], a co-designed intervention to improve communication about the heart failure trajectory [44], and co-designing complex interventions with people living with dementia and their carers [45]. Whether it is used for research or service improvement, co-design has been proved to be beneficial for study projects, users and services [40,43–48]. There is currently no evidence of co-design in the field of sarcopenia research, making the exploration of this study particularly valuable.

## Study aims and objectives

The overall aim of this research is to develop a theory-based and social-media based intervention for preventing possible sarcopenia through behaviour change in community-dwelling young-older adults. More specifically, the main objectives are: 1) To co-design a multicomponent intervention (health education plus exercise) strategy based on the SHEEP Conceptual Model; 2) To conduct a pre-test stage to refine both the intervention strategy and the SHEEP Conceptual Model. Thus, the following research questions will be posed: 1) What intervention contents should be included in different part of health education and exercise training? 2) How to ensure the correct dose of health education and exercise (in terms of frequency, intensity, and duration) can be achieved using a social-media based approach?

## Materials and methods

### Study design

The SHEEP programme was structured under the guidance of the UK Medical Research Council (MRC) for developing and evaluating complex interventions [49]. This research protocol outlined the initial stage (intervention development) in the MRC's guidance. Co-design method is the most crucial technique for completing the formulation of a social-media based intervention strategy to prevent sarcopenia among targeted population in this project. According to the five guiding principles and systematic framework proposed by Leask et al. [50] for designing, implementing, and evaluating co-designed public health interventions, consideration is given to first completing the designing phase in this study, in order to formulate a social-media based intervention strategy that incorporates the perspectives of the relevant stakeholders. This includes the preparing, developing, and pre-test phases (Fig 2).

· Preparing: In January and February of 2023, the research team developed an initial intervention plan based on literature reviews, the SHEEP Conceptual Model, and consultations with specialists regarding health education and exercise contents, frequency, intensity, and duration.

· Developing: Between March and June of 2023, there were two rounds of focus groups in the development procedure. Each round included three focus groups, to which older adults with possible sarcopenia were invited. The aim of the first round of focus groups was to discuss the initial intervention plan's flaws with participants and adapt it to the needs of the target population; we then created short videos based on the revised plan. The purpose of the second round of focus groups was to modify the content of each short video further.

· Pre-test: During July and August of 2023, a pre-test process was conducted with the goal of refining the SHEEP conceptual model and optimising the social-media-based intervention strategy and intervention process.

### Research setting

The entire study will be conducted at the Community Nursing Department of Xiang Ya Nursing School, Central South University, or within their cooperative community in

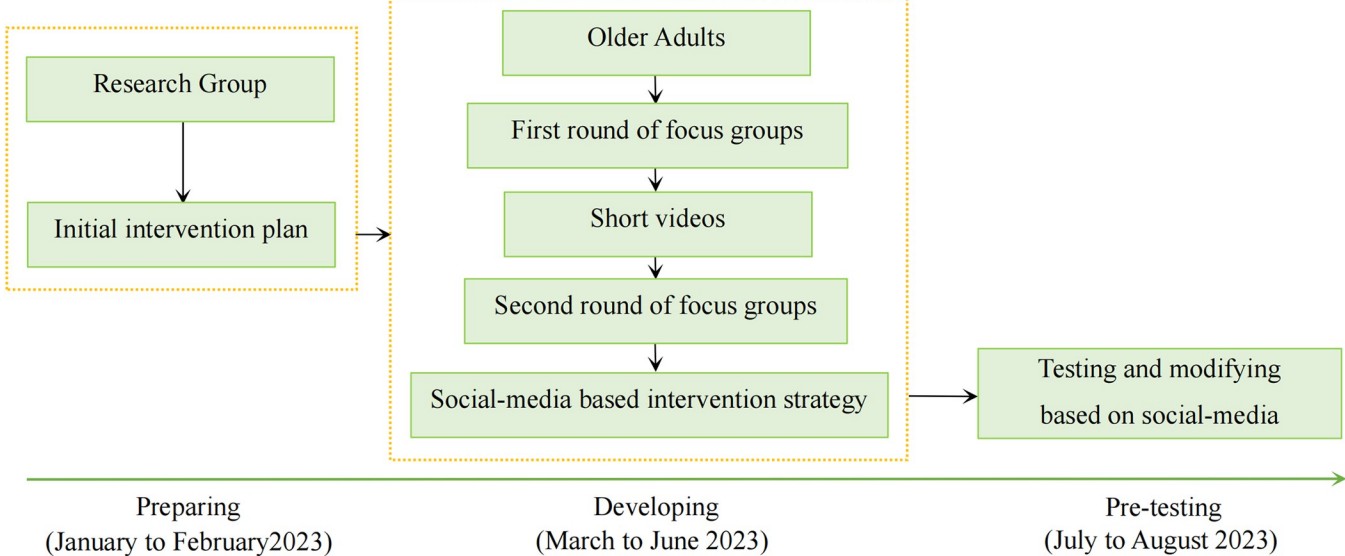

**Fig 2. Flow chart of this research protocol.** The study encompasses three main phases: preparation, development, and pre-test, which was plan to be completed during January to August, 2023.

Changsha, Human Province, China. It should be noted that the recruitment process will take place in a location that participants prefer based on their individual needs, such as the home of older participants or an office provided by our collaborator at Xiang Ya Nursing School or in the community. In addition, the co-design process will take place in an office provided by Xiang Ya Nursing School or the community, depending on the preference of the majority of participants within each focus group. However, if face-to-face focus groups are prohibited by local policy due to an outbreak, we will switch to online focus groups using Tencent Conference Software.

## Research participants

Although different scholars interpreted the definition of "young-old" differently, such as 60–69 years or 65–74 years [51,52], based on the fact that this study will be conducted in China, 60–69 years is considered young-old for the purposes of this study. Participants must satisfy the following inclusion criteria: (1) 60~69 years old; (2) Chinese residing in the community; (3) using TikTok prior to recruitment; (4) individuals with possible sarcopenia, as defined by low Grip Strength [M:<28 kg, F:<18kg] [2], which will be measured with a digital handheld dynamometer (EH101, Xiangshan Inc, Guangdong, China); (5) informed consent to screening and research. The following are the criteria for exclusion: (1) unable to communicate or independently complete learning on TikTok in Chinese; (2) with serious or unstable medical illness, such as severe cardiovascular or respiratory conditions, mental disorder, dementia, etc.; (3) sufficient physical activity ($\geq$ 150mins/week) [53,54].

## Sampling method and sample size

Purposive sampling will be used to select relevant stakeholders. To avoid bias against subgroups, we will select older adults based on their gender (male and female) and age (60~65y and 66~69y). We aim to recruit a maximum sample of 45 older adults in total. For developing phase, 18~25 older adults will be recruited to participate in focus groups (with a 10~15% attrition rate), and then invited to take part in both the first and the second round of focus groups. Each round will consist of three focus groups, with 18~25 participants divided into three groups of at least six members each, which is based on the recommendations of Leask et al. [50] that 10~12 co-creators in total is advised in co-design research and the Guidelines for Conducting a Focus Group that suggests six to ten people will be appropriate for a focus group [55]. Besides, we require 15~20 additional participants for the pre-test phase. In the two phases of our study, participants have the option of participating in only focus groups, only pre-test, or both focus groups and pre-test.

## Recruitment, consent and withdrawal

With the assistance of the cooperative community health centre's medical staff, we will first employ a population-based method to identify suitable subjects. First, we will visit a community health centre and inform health professionals about the case finding method, in which older individuals with the following clinical conditions are susceptible to possible sarcopenia: functional decline or limitation; unintentional weight loss; depressive mood; repeated falls; malnutrition; chronic conditions like heart failure, chronic obstructive pulmonary disease, diabetes mellitus, chronic kidney disease, etc. Second, we will only request that health professionals identify potential participants, not that they provide us with access to their medical records. We will provide health professionals with recruitment leaflets, and they will assist us in distributing the leaflets to potential population who visit the community health centre. Third, we will also send electronic recruitment leaflets to community WeChat groups

through the assistance of health professionals. If potential participants are interested in this study after reviewing the leaflet, they can contact the researcher via the email address or telephone number listed on the leaflet. Researchers will meet with older adults in their homes or at the community health centre, based on their preference, and then present our research in depth in person. If they are truly interested, we will implement a screening assessment to ensure they meet the aforementioned eligibility requirements.

Potential participants who meet the eligibility requirements will be provided with a Full Explanation of Research that informs them of the detailed research content, the confidentiality of personal information, the potential benefits, the potential risks, and a coping strategy for the risks, should they wish to know. A Participant Information Sheet will be provided to subjects who are still interested and willing to participate in the study, and time will be allowed for consideration (at least 24 hours). Those who agree to participate in the study will be required to provide written or verbal informed consent prior to data collection. If participants choose verbal consent, researchers will read each statement of the consent form and request their response via a dedicated mobile phone. Verbal consent will be separately recorded and stored.

Participants will be permitted to withdraw at any time without providing a reason. If participants have already participated in the focus group or in the pre-test phase, their contribution will not be retracted as it contributes to a broader discussion; however, no excerpts from their contributions during the discussions will be used in any report or presentation. Besides, researchers will evaluate participants' capacity to participate in the study and complete the intervention; if they are deemed incapable of providing consent, they will be withdrawn from the study, but the data already collected will be retained.

## Research ethics

This study has been approved by the University of Manchester Research Ethics Committee (Project ID: 15664), and permissions have already been granted by collaborators in the Community Nursing Department of Xiang Ya Nursing School, Central South University, China.

## Study procedures

We will adhere to the five key principles and systematic framework for co-designed public health interventions proposed by Leask et al. [50] throughout the entire research procedure. The five key principles are defining the purpose of the study, sampling, manifesting ownership, defining the procedure, and evaluating (the process and the intervention), all of which were structured into multiple sections to provide a systematic framework for the iterative co-creation process, including planning, conducting, reflecting, evaluating, and reporting (S3 Fig) [50]. As the study's sole objective is intervention development, we have modified our study procedure into three phases: preparing, developing, and pre-test, while still adhering to the aforementioned rules of principles and systematic framework.

· **Preparing phase (initial intervention content, frequency and duration).**   Based on previous literatures and the SHEEP conceptual model, we have formulated an initial intervention plan that includes health education and exercise, which will be modified in January and February 2023 after consultation with specialists in the field of sarcopenia and exercise for older adults.

Regarding health education, the curriculum would cover nine topics: What exactly is sarcopenia? What is sarcopenia's prevalence in older adults? What are the negative effects of sarcopenia? What factors contribute to sarcopenia? What clinical symptoms are associated with sarcopenia? What should you do if you suspect sarcopenia? What effects does physical

activity have? What physical activities can help prevent sarcopenia in older adults? What other evidence exists for preventing sarcopenia in older adults living in the community? The contents will be modified and expanded in the future, and will also be presented in short videos with language that Chinese older adults can easily comprehend, following the development of focus groups. Regarding intervention frequency, our scoping review found that 68.2% of health education groups for sarcopenia prevention recommended education frequency less than once per week [10]. This was relatively short and not equal to the frequency of physical activity (2–3 times/week) and/or nutrition groups (per day) [10], while this seems somewhat imbalanced. In order to facilitate comparison with the exercise group in future RCT trials, we initially set the frequency of health education at 3 times/ week. In addition, the duration per video is set to less than 5 minutes to aid the memory of older individuals. However, these parameters may be modified after the development and pre-test phases.

In terms of exercise, resistance exercise (RE) will be the primary form of training for sarcopenia prevention, but specific motions must be selected and adapted in the co-design stage based on the characteristics of the Chinese older population. Systematic review and meta-analysis provided robust evidence that RE programmes can improve muscle mass and strength in middle-aged and older adults [56,57]. Aside from RE, we also consider a combination of other exercise modes including balance training (BT) and aerobic training (AT), as older adults with sarcopenia are also likely to be at an increased risk for falls [58] and exhibit reduced cardiorespiratory fitness [59]. It should be noted that there are currently no specific RE guidelines for possible sarcopenia; therefore, we will refer to the individualised prescription of RE for older adults with sarcopenia (Table 1) [60], which will be modified during the course of this study to make it more feasible for the target population. Frequency and duration of exercise are progressive from the beginning to the end of the intervention, with frequency increasing from 3 days/week to 4 days/week and duration increasing from 20mins/session to 40mins/session; as the National Health Service (NHS) and Centres for Disease Control and Prevention (CDC) recommend older individuals at least 2 days/week and 150mins/week of activities to strengthen muscles [53,54]. Similarly with health education, all exercise components will be refined and produced as short videos following the development phase. Hence, the initial multicomponent exercise strategy contains warm-up training, aerobic training, balance training, resistance training, and flexibility training to cool down (Table 2).

· **Development phase.** The development stage mainly includes two rounds of consultation with patient and public involvement (PPI) members and two rounds of focus groups to facilitate the development of an acceptable SHEEP intervention strategy for the targeted older population.

**Table 1. RE prescription of exercise for older adults with sarcopenia, Hurst et al. 2022 [60].**

| Dimensions | Criteria |
|---|---|
| Principles of exercise training | specificity, overload and progression |
| Training frequency | two sessions per week |
| Exercise selection | lower body (Squat/leg press, Knee extension, leg curl, calf raise), upper body (chest press, seated row, pull down) |
| Exercise intensity | repetition-continuum based prescription (40~60% 1RM progressing to 70~85% 1RM), Rating of Perceived Exertion (RPE)-based prescription (RPE 3~5 on Category Ratio 10 scale progressing to RPE 6~8) |
| Exercise volume | 1~3 sets of 6~12 repetitions |
| Rest periods | within session (60~120s between sets and 3~5mins between exercises), between sessions (at least 48 h) |

**Table 2. Example of exercises for different exercise type.**

| Exercise type | Example of exercises |
|---|---|
| Warm-up training | marching, side taps, head circles, shoulder lift, shoulder circles, side bends, trunk twist |
| Aerobic training | marches and bigger, side steps, double side steps, high-fives under crotch, knee lifts with hands down, knee lifts with punch first, front lunges, simple jumping jacks |
| Balance training | Flamingo swing, heel walking, Tandem walking, upward reach balance |
| Resistance training | sit to stand strength, outer thigh strength standing, stride knee bends strength, wrist strength, front arm strength, back arm strength, wall press up |
| Flexibility training | back of upper arm stretch, upwards side stretch, inner thigh stretch, back of thigh stretch |

The first round of PPI consultation will be conducted face-to-face with six to eight young-old seniors between the ages of 60 and 69, who will be selected using convenience sampling methods. The purpose of this consultation is to gain preliminary understanding of the ability of Chinese young-old adults to understand and undertake the intervention; and to solicit their feedback on the content and duration of the initial SHEEP health education. Principal participants in the second round of PPI consultation will include a gerontologist, a geriatric nurse, a rehabilitation physician, and a rehabilitation nurse from Xiangya Nursing School and Xiangya Second Affiliated Hospital, Central South University, China. The gerontologist and geriatric nurse should have rich experience of geriatric health education, and their consultation is intended to make the SHEEP health education strategy more systematic, detailed, and in line with lay language requirements for Chinese older adults. Likewise, the purpose of the consultation with the rehabilitation physician and nurse with deep knowledge of geriatric sports medicine is to revise the SHEEP exercise intervention to ensure its safety and suitability for older people.

Using topic guides, at least two rounds of focus groups comprised of older adults with possible sarcopenia will be conducted (Fig 3). There will be at least six older adults in each focus group during each round. The optimal time allotted for each focus group is 45 to 90 minutes, during which no more than twelve predetermined questions are discussed [55]. If face-to-face focus groups are prohibited by local epidemic policy, we will consider conducting online focus groups instead. The purpose of the first round of focus groups is to discuss the problems in the paper version of the initial intervention plan and modify it based on the suggestions of our participants to better fit the targeted population. Following this, we will create short videos based on the revised paper plan. The short videos will include two categories of short instructional videos for TikTok: health education and exercise (Fig 4). Then, the second round of focus groups will seek to modify the content of each short video further. If there are numerous issues with short videos, we will consider conducting a third round of focus groups to review all of the revised short videos and ensure that each segment is appropriate for our stakeholders and free of controversy.

·**Pre-test phase.** We will then obtain a fully theory-based conceptual model and a social-media based intervention strategy, which will be tested and modified in a small pre-feasibility study during three weeks. One week prior to pre-test, we will conduct a training session to instruct participants on how to follow our TikTok account, find our short videos, and provide us with feedback during the pre-test phase. Following the commencement of pre-test, we will distribute nine health education videos over the course of three weeks, three short videos per week; concurrently, we will schedule weekly comprehensive exercise videos, including balance training, aerobic training, and resistance training videos. The primary observations and modifications pertain to the conceptual model, intervention strategy, and intervention

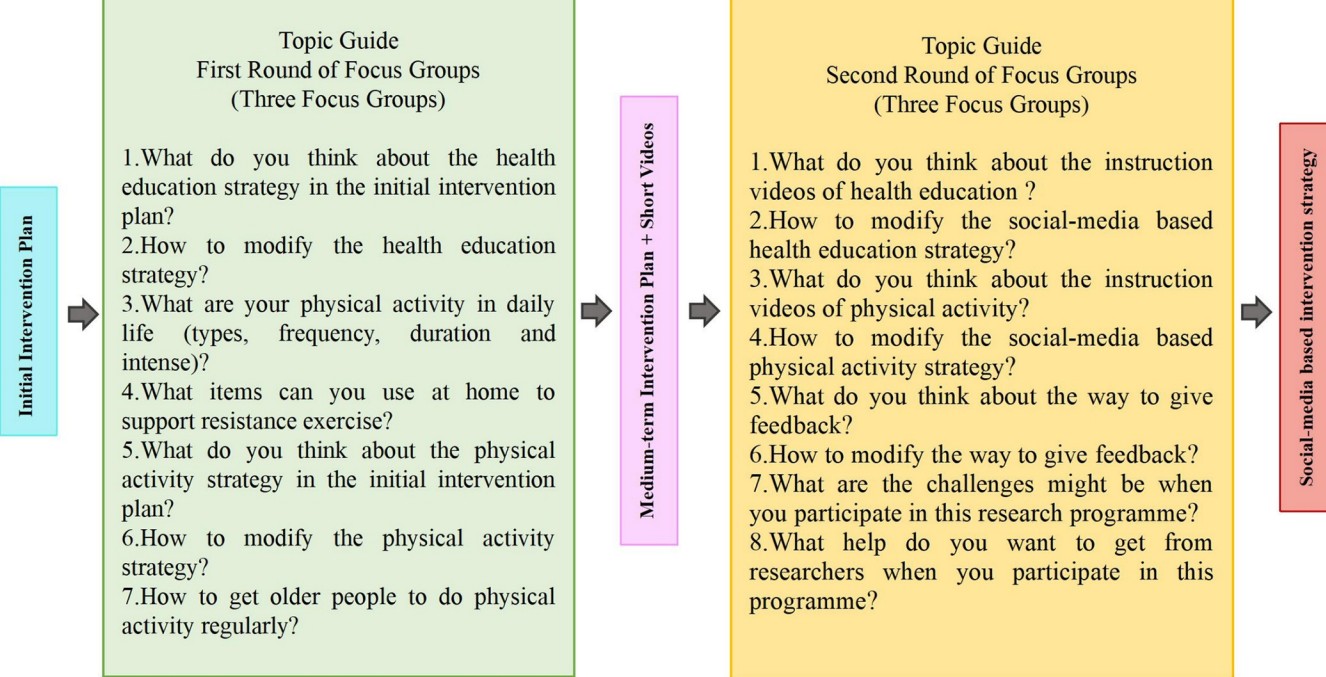

**Fig 3. Topic guide of two rounds of focus groups.** The topic guide described in this diagram is just a preliminary idea, and the specific details was plan be presented in the real practice stage.

process, such as the researchers' guiding process and older adults' learning process. We will use success criteria (Fig 5) to determine whether to proceed to the subsequent phase. Each of the success criteria will be graded on a scale from one to five: completely substandard, a small portion of the standard, fifty percent of the standard, the majority of the standard, and completely up to standard. We will establish the pass criterion as meeting more than 80% of the total success criteria; if not, we will revise it to enter the next round of three-week pre-test; if it is met, we will refine it and enter the next phase, a feasibility study.

We will mainly collect two categories of data from participants, including intervention compliance and centralized feedback, in order to verify the appropriate contents and the correct dose of health education and exercise (in terms of frequency, intensity, and duration) can be achieved using a social-media-based approach.

1. Intervention compliance: Depending on their preference, participants can inform researchers of their completion of viewing instructional videos and physical activity via phone call or text message. To monitor participant compliance, the researcher will keep a three-week diary for each participant using a dedicated mobile phone that is not a personal own. The researcher will purchase a new SIM card and dedicated phone number in China for this study.

2. Centralized feedback: This will be a semi-structured interview conducted face-to-face or over the phone in order to delve into the essence of problems and solicit suggestions for future revision. The semi-structured questions would be like: How do you feel throughout the entire pre-test phase? What do you think about the social media-based health education in pre-test phase? How to improve the social- media based health education? What do you think about the social-media based physical activity in pre-test phase? How to improve the

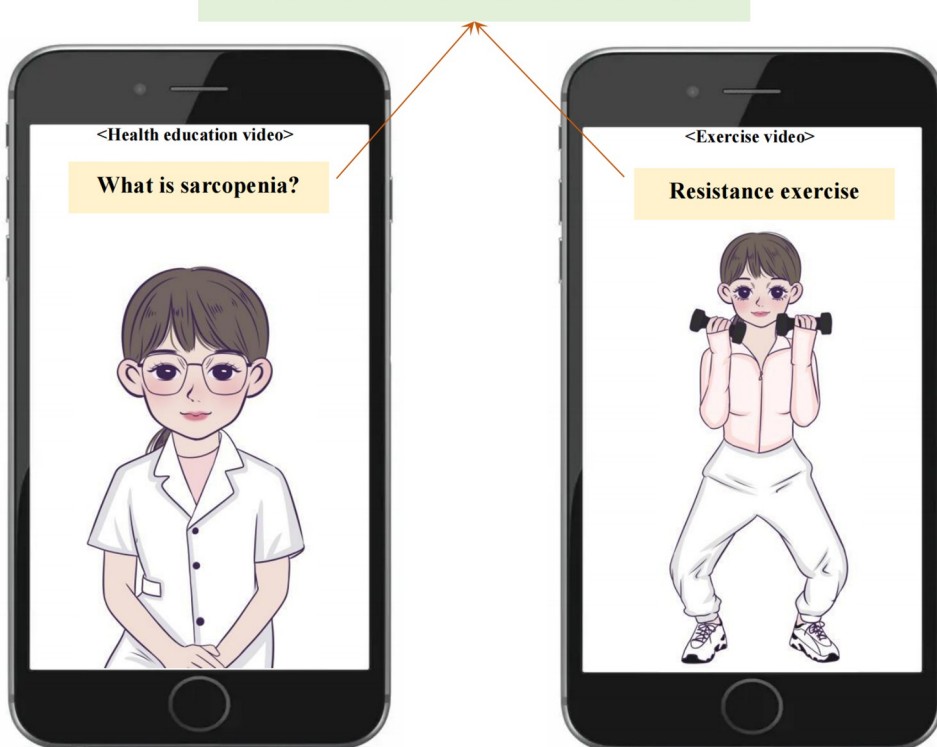

**Fig 4. Instructional video examples for TikTok.** The instruction videos can be divided into two categories, including health education videos and exercise videos. The animation figure shown here is merely a preliminary example, and the actual video content was plan to be shown after the study is complete.

social-media based physical activity? In addition to health education and physical activity, what other aspects of this project do you believe require improvement? What do you believe the challenges will be if this programme is extended to a greater number of older individuals with possible muscle loss?

Moreover, researchers should document issues that arise during the guiding process, such as uploading videos to TikTok, distributing notices in TikTok groups, monitoring and recording compliance of behaviour change, and determining whether communication on TikTok is efficient.

## Data collection and analysis

To promote the collection of high-quality data, the assessors will double-check the collected data, and a statistician will examine the range of values. All data will be saved on servers or computers that are certified by the University of Manchester and have safe backup systems. The data will be analyzed in China, in accordance with the data management plan.

As for quantitative data analysis, descriptive statistics will be utilised to summarise the percentage of recruitment, baseline participant characteristics (e.g. age, gender, and educational background), attrition rate, compliance rate, etc. All statistical analyses will be conducted using SPSS Statistics 27.0 (IBM Corp., Armonk, NY, USA). P Values <0.05 will be interpreted as indicating significance.

Regarding the collection of qualitative data, all face-to-face recordings of group interviews in focus groups and final semi-structured interviews in the pre-test phase will be audio

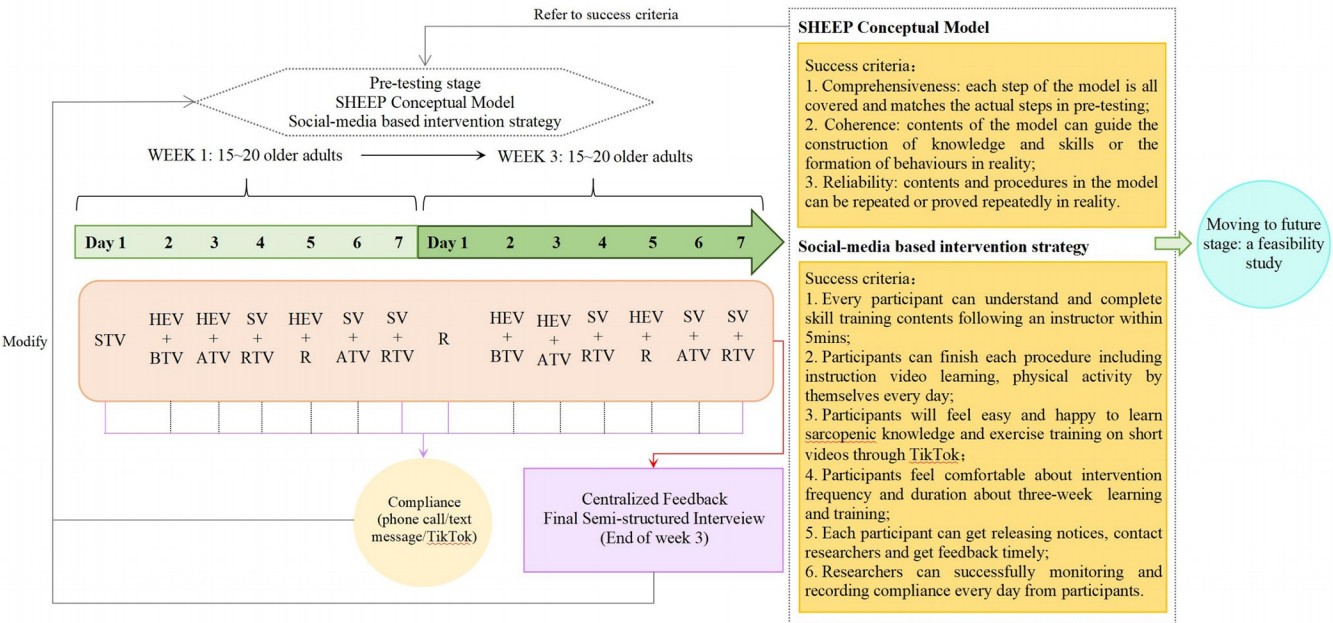

**Fig 5. A brief diagram of pre-test procedure.** The intervention strategy in this diagram is just an example using for demonstration, which was plan to be modified and refined during the study period. The complete designations of the abbreviations in this diagram are as follows: STV, skill training video; HEV, health education video; BTV, balance training video; ATV, aerobic training video; RTV, resistance training video; SV, summary video; R, rest.

recorded using an encrypted digital recorder. If face-to-face meetings are prohibited by local epidemic policy, the focus group will be conducted using Tencent Conference Software, which can encrypt meetings to protect participants' privacy, and only audio of the group meeting will be captured via the cloud recording function. In addition, if participants prefer a telephone interview for the final semi-structured interview, the conversation will be recorded on an encrypted dictaphone. All encrypted recordings will be uploaded to a secure server at the University of Manchester and then deleted from the recording device and Tencent Conference cloud. Moreover, each interview will last between 30 minutes and one hour, and participants will only be asked to participate in one.

During the data analysis stage, qualitative data will be transcribed verbatim and potentially identifying information will be removed. The transcripts will be compared to the original recordings, corrected as needed, and anonymized. To become familiar with qualitative data, researchers must first read all transcripts. Themes will be developed based on the interview guide and research team discussion in order to identify the key characteristics of the qualitative data. Then, codes will be generated for each line of each transcript and categorised accordingly. Before finalising the major themes, each category must be re-evaluated and scrutinised, and groups of major categories will be refined. Two researchers will perform and review the coding process to ensure a double check, followed by a discussion to ensure the validity of the data. To increase the transparency of the interpretation, the Chinese-to-English and English-to-Chinese translation of quotations will be performed in this study. QSR International's NVivo 12 qualitative analysis software will be used to assist and facilitate the coding and analysis process.

## Trial monitoring

The principal investigator will oversee participant recruitment, intervention development and pre-test stage. In addition, one community leader in the cooperative community health centre

will assume overall responsibility for participant identification, recruitment, and pre-test. One team comprised of research professionals and academic experts from the United Kingdom and China will be responsible for ensuring the overall quality of research data. All principal researchers will hold a monthly online meeting to report on the project's progress and discuss its problems and solutions. A thorough risk assessment has been conducted, and potential patient, organisational, and study hazards, as well as their likelihood of occurrence and potential consequences, have been considered.

## Discussion

We have presented a protocol for developing a theory-based and social-media based multicomponent intervention (health education + exercise) with the co-design method in community-dwelling young-older adults to prevent possible sarcopenia. This research project was inspired by the opportunities afforded by the development of digital technologies. The World Health Organization announced a Global Strategy on Digital Health 2020–2024 with the mission to "improve health for everyone, everywhere by accelerating the adoption of appropriate digital health [61]." Thus, digital health has the potential to be a paradigm shifter in terms of enhancing information, education, communication, health monitoring, diagnostics, and data management [62]. There are few internet-based interventions for sarcopenia prevention. One study in China involved a 12-week intervention of 234 older adults with sarcopenia and revealed that nutrition interventions through an app were effective in optimising the diets and increasing the skeletal muscle mass [63]. Another study conducted a randomized controlled trial on 23 participants and indicated that supervised resistance exercise through a video conferencing software (Skype™) had positive effects on sarcopenia-related factors among community-dwelling senior citizens in South Korea [64]. However, social media-based interventions to prevent sarcopenia have not yet been reported, so this project is novel. The SHEEP programme was conceived against such a backdrop.

This project has some strengths. First of all, SHEEP has considerable potential as it can meet the needs of the information age, providing targeted timely intervention to older people through social media. This offers numerous advantages in terms of accessibility, time and cost savings, and continuity of care. In addition, SHEEP was structured under the guidance of the MRC for developing and evaluating complex interventions [49], and the sound construction method could enhance the repeatability of the study. Moreover, the intervention's content and process were developed based on a theoretical framework, thereby enhancing its credibility.

This research has several limitations. First, as the proposed study focuses solely on intervention development, the sample size will be small; consequently, it may not be representative of the entire Chinese older population. Second, older adults who are unfamiliar with TikTok or lack Internet and modern communication technology access or proficiency were excluded from the recruitment stage, which may result in a less diverse sample. Third, because of the short duration of the pre-test phase, we will not draw any conclusions about the intervention effectiveness, which will be evaluated in a subsequent study. In addition, the study findings are unique to the community setting and population in China and may not be generalised to other settings or countries. Nevertheless, this study will demonstrate whether the social-media based intervention content, frequency, duration, process are appropriate to young-older adults with possible sarcopenia.

Finally, the findings from the two rounds of focus groups in this programme will be a SHEEP conceptual model and a multicomponent intervention strategy, which will not only effectively compensate for the deficiencies of health education in the field of sarcopenia, but will also provide a theoretical framework for the transformation from knowledge to behaviour

change of sarcopenia intervention based on social media, with the aim of establishing a theoretical foundation for sarcopenia intervention development. In addition, the results of observation and interviews in this study will provide researchers with information on the habits of Chinese older adults who use social media for health education learning and exercise. Furthermore, the comprehensive co-design procedure will build a foundation for co-design research in the field of sarcopenia. Additionally, the health education and exercise video resources will serve as valuable reference materials for intervention by other researchers. All of these findings will serve as evidence for our future feasibility study and will be disseminated via publication in peer-reviewed journals and presentation at relevant conferences, and all relevant data will be made available within the manuscript and its Supporting Information files, in order to serve as valuable resources, references, and directions for other researchers in related fields.

## Supporting information

**S1 Fig. Behaviour change wheel.**
(TIF)

**S2 Fig. eLiFE conceptual model.**
(TIF)

**S3 Fig. Iterative co-creation process.**
(TIF)

## Author Contributions

**Conceptualization:** Ya Shi, Emma Stanmore, Lisa McGarrigle, Chris Todd.

**Data curation:** Ya Shi.

**Formal analysis:** Ya Shi.

**Funding acquisition:** Ya Shi, Chris Todd.

**Investigation:** Ya Shi.

**Methodology:** Ya Shi, Emma Stanmore, Lisa McGarrigle, Chris Todd.

**Project administration:** Ya Shi, Emma Stanmore, Lisa McGarrigle, Chris Todd.

**Resources:** Ya Shi, Emma Stanmore, Lisa McGarrigle, Chris Todd.

**Software:** Ya Shi.

**Supervision:** Emma Stanmore, Lisa McGarrigle, Chris Todd.

**Validation:** Emma Stanmore, Lisa McGarrigle, Chris Todd.

**Visualization:** Ya Shi.

**Writing – original draft:** Ya Shi.

**Writing – review & editing:** Emma Stanmore, Lisa McGarrigle, Chris Todd.

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
