## [Decision Letter · Decision Letter 0]

9 Jul 2023

PONE-D-23-14790Social-media based Health Education plus Exercise Programme (SHEEP) to improve muscle function among young-old adults with possible sarcopenia in the community a study protocol for intervention developmentPLOS ONE

Dear Dr. SHI,

Thank you for submitting your manuscript to PLOS ONE. After careful consideration, we feel that it has merit but does not fully meet PLOS ONE’s publication criteria as it currently stands. Therefore, we invite you to submit a revised version of the manuscript that addresses the points raised during the review process.

If you decide to revise the work, please submit a list of changes (or a rebuttal against each point raised) along with the revised manuscript.

We look forward to receiving your revised manuscript.

Kind regards,

Sayani Das, MPhil

Academic Editor

PLOS ONE

- https://www.sciencedirect.com/science/article/pii/S0167494323001000?via%3Dihub

https://bmjopen.bmj.com/content/bmjopen/13/2/e067079.full.pdf

https://link.springer.com/article/10.1007/s13312-020-1789-7

In your revision ensure you cite all your sources (including your own works), and quote or rephrase any duplicated text outside the methods section. Further consideration is dependent on these concerns being addressed.

6. Please include a separate caption for each figure in your manuscript.

7. We note that Figure 7 in your submission contain copyrighted images. All PLOS content is published under the Creative Commons Attribution License (CC BY 4.0), which means that the manuscript, images, and Supporting Information files will be freely available online, and any third party is permitted to access, download, copy, distribute, and use these materials in any way, even commercially, with proper attribution. For more information, see our copyright guidelines: http://journals.plos.org/plosone/s/licenses-and-copyright.

a. You may seek permission from the original copyright holder of Figure 7 to publish the content specifically under the CC BY 4.0 license.

b.If you are unable to obtain permission from the original copyright holder to publish these figures under the CC BY 4.0 license or if the copyright holder’s requirements are incompatible with the CC BY 4.0 license, please either i) remove the figure or ii) supply a replacement figure that complies with the CC BY 4.0 license. Please check copyright information on all replacement figures and update the figure caption with source information. If applicable, please specify in the figure caption text when a figure is similar but not identical to the original image and is therefore for illustrative purposes only.

Additional Editor Comments (if provided):

1. All the components (mainly introduction section) need to be concise and focused as they are currently too lengthy. Kindly condense them appropriately.

2. The Discussion section requires further development. It should begin by providing a concise summary of the main findings, followed by situating them within the existing knowledge base, including relevant theories and empirical literature. Furthermore, it should comprehensively outline the implications of the findings for future research, policy-making, and practical applications.

3. Please provide DOIs for all the references.

Reviewers' comments:

Reviewer's Responses to Questions

**Comments to the Author**

1. Does the manuscript provide a valid rationale for the proposed study, with clearly identified and justified research questions?

Reviewer #1: Yes

Reviewer #2: Yes

2. Is the protocol technically sound and planned in a manner that will lead to a meaningful outcome and allow testing the stated hypotheses?

Reviewer #1: Yes

Reviewer #2: Yes

3. Is the methodology feasible and described in sufficient detail to allow the work to be replicable?

Reviewer #1: Yes

Reviewer #2: Yes

4. Have the authors described where all data underlying the findings will be made available when the study is complete?

Reviewer #1: Yes

Reviewer #2: No

5. Is the manuscript presented in an intelligible fashion and written in standard English?

Reviewer #1: Yes

Reviewer #2: Yes

6. Review Comments to the Author

You may also provide optional suggestions and comments to authors that they might find helpful in planning their study.

Reviewer #1: Very well written, just a conclusion needs to be added explaining the future scope of research and how other scholars would benefit from this research.

Reviewer #2: I appreciate the opportunity to review your manuscript titled “Social-media based Health Education plus Exercise Programme (SHEEP) to improve muscle function among young-old adults with possible sarcopenia in the community a study protocol for intervention development". I would like to commend you for your efforts in researching this topic and for the valuable insights presented in the manuscript. Overall, I believe your study has the potential to make a significant contribution to the field of gerontology. However, there are a few areas that require attention and improvement. I have outlined my comments below (also specific suggestions and queries are annotated in a separate pdf file):

Your introduction provides a clear overview of the research problem and its significance. However, I recommend providing specific definition of possible sarcopenia with proper reference to make it more understandable to the readers. Also, I suggest to elaborate the existing literature that have cited, especially the population and area where the study have conducted to better contextualize the study. Emphasise on the previous studies that address social media-based intervention, digital health for older adults etc. This will strengthen the background and rationale for your research.

The ‘Methods and Analysis’ section is well structured, though some sub-sections required more elaboration to better understand your work as well as to replicate this protocol in similar future studies.

The discussion effectively summarizes the key aspects of this study and its requirement in recent decades. However, I encourage you to consider incorporating few relevant studies in the same vein to reflect your scope of plan with others.

Overall, I find your manuscript to be significant and worth considering for publication after addressing the suggestions. I believe that by incorporating these suggestions, your work will significantly enhance its clarity and impact on the field.

7. PLOS authors have the option to publish the peer review history of their article (what does this mean?). If published, this will include your full peer review and any attached files.

Reviewer #1: **Yes: **Jagriti Gangopadhyay

Reviewer #2: No

---

## [Author Response · Author response to Decision Letter 0]

4 Aug 2023

Dear Editor and Reviewers: 

Manuscript ID number: PONE-D-23-14790

Title: Social-media based Health Education plus Exercise Programme (SHEEP) to improve muscle function among young-old adults with possible sarcopenia in the community: a study protocol for intervention development

Thank you so much for your insightful feedback on our manuscript. We have revised and responded to each comment in accordance with your suggestion, as demonstrated below.

Journal Requirements

We have modified our manuscript according to the requirements of two documents, PLOSOne_formatting_sample_main_body and PLOSOne_formatting_sample_title_authors_affiliation

2.We noticed you have some minor occurrence of overlapping text with the following previous publication(s), which needs to be addressed.

In your revision ensure you cite all your sources (including your own works), and quote or rephrase any duplicated text outside the methods section. Further consideration is dependent on these concerns being addressed.

We have cited all appropriate sources, including our own publications, and have rephrased or removed any duplicated text. (P5)

We have added a brief explanation in “recruitment, consent and withdrawal” section about when and how informed consent will be obtained from participants, as well as how it should be recorded if consent is given verbally. (P8-9)

We have modified the ‘Funding Information’ and ‘Financial Disclosure’ sections as required. (Cover Letter)

5.Your ethics statement should only appear in the Methods section of your manuscript. If your ethics statement is written in any section besides the Methods, please move it to the Methods section and delete it from any other section. Please ensure that your ethics statement is included in your manuscript, as the ethics statement entered into the online submission form will not be published alongside your manuscript.

We have moved the ethics statement to the Methods section. (P9)

6.Please include a separate caption for each figure in your manuscript.

We have captioned all the figures and saved them individually, and referenced them correctly according to the format requirements. (P16)

7.We note that Figure 7 in your submission contain copyrighted images. All PLOS content is published under the Creative Commons Attribution License (CC BY 4.0), which means that the manuscript, images, and Supporting Information files will be freely available online, and any third party is permitted to access, download, copy, distribute, and use these materials in any way, even commercially, with proper attribution.

We have substituted the cartoon images in Figure 7 with original images that were drawn by the first author. The two original cartoon characters are portraits of the first author. Therefore, the copyright of the images belongs to the first author, and she agrees to publish them under the Creative Commons Attribution License (CC BY 4.0).

8.Please review your reference list to ensure that it is complete and correct. If you have cited papers that have been retracted, please include the rationale for doing so in the manuscript text, or remove these references and replace them with relevant current references. Any changes to the reference list should be mentioned in the rebuttal letter that accompanies your revised manuscript. If you need to cite a retracted article, indicate the article’s retracted status in the References list and also include a citation and full reference for the retraction notice.

We have reviewed all the references to ensure their complete and correct. (P16-20)

Editor Comments

1.All the components (mainly introduction section) need to be concise and focused as they are currently too lengthy. Kindly condense them appropriately.

The whole paper including introduction, methods, discussion and references have been reduced from 7450 to 7100 words, especially, the introduction section has been reduced from 1460 to 1250 words. 

2.The Discussion section requires further development. It should begin by providing a concise summary of the main findings, followed by situating them within the existing knowledge base, including relevant theories and empirical literature. Furthermore, it should comprehensively outline the implications of the findings for future research, policy-making, and practical applications.

We have modified the discussion section per your insightful suggestion. The first paragraph provides a concise summary of the project, and a comparison to existing literature; the second paragraph describes the strengths of this project; the third paragraph outlines the limitations; and the fourth paragraph discusses the implications of the findings. (P14-15)

3.Please provide DOIs for all the references.

We have added DOIs for all the references. (P16-20)

Reviewers' Comments

Reviewer #1: Very well written, just a conclusion needs to be added explaining the future scope of research and how other scholars would benefit from this research.

Thank you so much for your valuable advice. We have referenced the publisher's recommendations for the protocol format and other publications in this journal, and then included the implications of the findings, the future research scope, and how other scholars can benefit from this study in the final paragraph. (P14-15)

Reviewer #2: I appreciate the opportunity to review your manuscript titled “Social-media based Health Education plus Exercise Programme (SHEEP) to improve muscle function among young-old adults with possible sarcopenia in the community a study protocol for intervention development". I would like to commend you for your efforts in researching this topic and for the valuable insights presented in the manuscript. Overall, I believe your study has the potential to make a significant contribution to the field of gerontology. However, there are a few areas that require attention and improvement. I have outlined my comments below (also specific suggestions and queries are annotated in a separate pdf file):

Your introduction provides a clear overview of the research problem and its significance. However, I recommend providing specific definition of possible sarcopenia with proper reference to make it more understandable to the readers. Also, I suggest to elaborate the existing literature that have cited, especially the population and area where the study have conducted to better contextualize the study. Emphasise on the previous studies that address social media-based intervention, digital health for older adults etc. This will strengthen the background and rationale for your research.

The ‘Methods and Analysis’ section is well structured, though some sub-sections required more elaboration to better understand your work as well as to replicate this protocol in similar future studies.

The discussion effectively summarizes the key aspects of this study and its requirement in recent decades. However, I encourage you to consider incorporating few relevant studies in the same vein to reflect your scope of plan with others.

Overall, I find your manuscript to be significant and worth considering for publication after addressing the suggestions. I believe that by incorporating these suggestions, your work will significantly enhance its clarity and impact on the field.

Thank you so much for your encouragement and patient guidance. We have carefully read each of your comments in the PDF file and made detailed modifications to the corresponding parts of the manuscript according to your suggestions.

Introduction

1.We have added the definition of possible sarcopenia at the beginning of the introduction section. (P4)

2.We have supplemented all cited studies with corresponding populations and regions in order to facilitate research status comparisons between regions. (P4)

3.We have detailed the findings from our scoping review that were instructive for this study. (P4-5)

4.We have listed several literatures to show that social media may be a promising medium for health education to disseminate knowledge and raise awareness of disease prevention among older adults. (P5)

Methods and Analysis

1.We have specified that the two rounds of focus groups will be conducted with the same participants, and that each round will consist of three groups, and that the participants will be distributed evenly among the three groups. (P8)

2.We have explained the reason that we contemplated setting the frequency of health education at 3 times/week. (P10)

3.To serve as a better resource for other scholars, we have added the initial multicomponent exercise strategy's detailed contents. (P11)

4.We have specified that if 80% of the total success criteria are met during pre-testing, the feasibility study will be initiated; otherwise, it will be modified to require an additional three weeks of pre-testing. (P12)

Discussion

On the basis of your advice, we have added several additional studies to elucidate current internet-based interventions for preventing sarcopenia in order to facilitate a more accurate comparison with this study. (P14-15)

Finally, we really appreciate all the time and effort you put into this document, as well as all your insightful suggestions for revising the manuscript to make it more readable, clear, and academic. Thank you so much once more!

Best wishes

---

## [Decision Letter · Decision Letter 1]

13 Oct 2023

PONE-D-23-14790R1Social-media based Health Education plus Exercise Programme (SHEEP) to improve muscle function among young-old adults with possible sarcopenia in the community a study protocol for intervention developmentPLOS ONE

Dear Dr. SHI,

Thank you for submitting your manuscript to PLOS ONE. After careful consideration, we feel that it has merit but does not fully meet PLOS ONE’s publication criteria as it currently stands. Therefore, we invite you to submit a revised version of the manuscript that addresses the points raised during the review process. We have completed the review of your manuscript and a summary is appended below. The reviewer(s) have recommended some minor revisions to your manuscript.  Therefore, I invite you to respond to the reviewer(s)' comments and revise your manuscript.

We look forward to receiving your revised manuscript.

Kind regards,

Dr Sayani Das, PhD

Academic Editor

PLOS ONE

Journal Requirements:

Reviewers' comments:

Reviewer's Responses to Questions

**Comments to the Author**

1. Does the manuscript provide a valid rationale for the proposed study, with clearly identified and justified research questions?

Reviewer #2: Yes

Reviewer #3: Yes

2. Is the protocol technically sound and planned in a manner that will lead to a meaningful outcome and allow testing the stated hypotheses?

Reviewer #2: Yes

Reviewer #3: Yes

3. Is the methodology feasible and described in sufficient detail to allow the work to be replicable?

Reviewer #2: Yes

Reviewer #3: No

4. Have the authors described where all data underlying the findings will be made available when the study is complete?

Reviewer #2: Yes

Reviewer #3: No

5. Is the manuscript presented in an intelligible fashion and written in standard English?

Reviewer #2: Yes

Reviewer #3: Yes

6. Review Comments to the Author

You may also provide optional suggestions and comments to authors that they might find helpful in planning their study.

Reviewer #2: Appreciate your effort and dedication to addressing the comments/suggestions in improving this manuscript. The study intervention protocol is well described in methods section. Intervention studies in gerontology is utmost required to mitigate the burden of population ageing and to provide good health for older adults.

However, the manuscript required few minor corrections (highlighted in the pdf document).

Reviewer #3: This a a well written protocol. However, some points need clarification

a. Sample size calculation

b. Literacy and social media use of the population studied, If the study protocol personnel want to know Tik Tok use among the population, they could use a better topic, which would be more interesting

c. The setting is a population covered by a Nursing School - who are the content experts in this field who will vet the data and create the health education programs

d. Please clearly define which criteria you are going to use to define possible sarcopenia

e. Addressing possible sarcopenia is a multipronged approach. One intervention - that too unsupervised will not make much of a difference.

f. What are the end points - who will be the judge

7. PLOS authors have the option to publish the peer review history of their article (what does this mean?). If published, this will include your full peer review and any attached files.

Reviewer #2: **Yes: **Dr. Anushka Ghosh

Reviewer #3: No

---

## [Author Response · Author response to Decision Letter 1]

23 Nov 2023

Journal Requirements 

1.Please review your reference list to ensure that it is complete and correct. If you have cited papers that have been retracted, please include the rationale for doing so in the manuscript text, or remove these references and replace them with relevant current references. Any changes to the reference list should be mentioned in the rebuttal letter that accompanies your revised manuscript. If you need to cite a retracted article, indicate the article’s retracted status in the References list and also include a citation and full reference for the retraction notice. 

Thank you for this comment. We have reviewed and checked the reference list to ensure that it is complete and correct. (P17-21)

Reviewers' Comments

3.Is the methodology feasible and described in sufficient detail to allow the work to be replicable?

Reviewer #2: Yes

Reviewer #3: No 

We have added more details in methodology section about how to develop the intervention strategy, according to the reviewers’ suggestion. (P12)

4.Have the authors described where all data underlying the findings will be made available when the study is complete?

Reviewer #2: Yes

Reviewer #3: No 

All of these findings will serve as evidence for our future feasibility study and will be disseminated via publication in peer-reviewed journals and presentation at relevant conferences, in order to serve as valuable resources, references, and directions for other researchers in related fields. (P16)

6.Review Comments to the Author

Reviewer #2: Appreciate your effort and dedication to addressing the comments/suggestions in improving this manuscript. The study intervention protocol is well described in methods section. Intervention studies in gerontology is utmost required to mitigate the burden of population ageing and to provide good health for older adults. However, the manuscript required few minor corrections (highlighted in the pdf document).

Thank you so much for your valuable advice. We have carefully read each of your comments in the PDF file and made detailed modifications to the corresponding parts of the manuscript according to your suggestions.

1)We have included the year within parenthesis. (P4)

2)We have moved the definition of “young-old” in this study from the introduction to the methodology. (P4;P8)

3)We have revised the format of the literature citation. (P5)

4)We have rewritten relevant sentences in the past tense in the section of study design. (P7)

5)We have revised the format of the literature citation. (P8)

6)We have deleted an extra comma “,”. (P9)

7)We have rewritten the word “pre-paring” as “preparing”. (P10)

8)We have deleted a repetition “to”. (P11)

Reviewer #3: This a a well written protocol. However, some points need clarification

a.Sample size calculation

Since this study was designed to develop an intervention strategy, there was no formula for sample size calculation. We carefully referred to other similar studies on intervention development to estimate sample size, and we also provided relevant references. (P8)

b.Literacy and social media use of the population studied, If the study protocol personnel want to know Tik-Tok use among the population, they could use a better topic, which would be more interesting

Our goal is not to investigate the use of TikTok in the population. Our fundamental goal is to intervene in older adults with possible sarcopenia. Social media (TikTok) is just a tool for our intervention, and it is an attempt to combine the current trend of social media. The materials could be used in other social media platforms in the future if needed but our choice of TikTok for this study is due to its wide uptake and ability to engage with our study population. 

c. The setting is a population covered by a Nursing School - who are the content experts in this field who will vet the data and create the health education programs

The Xiang Ya Nursing School we mentioned is our research partner, and they will provide the necessary experts, research equipment and site support for our study, but they do not participate in the management and analysis of the research data. During the preparing stage, the initial intervention plan was just designed by the research team based on literature reviews and team meetings. 

We have added more details about how experts work in our study, as follows. (P12)

The development stage mainly includes two rounds of consultation with patient and public involvement (PPI) members and two rounds of focus groups to facilitate the development of an acceptable SHEEP intervention strategy for the targeted older population.

The first round of PPI consultation will be conducted face-to-face with six to eight young-old seniors between the ages of 60 and 69, who will be selected using convenience sampling methods. The purpose of this consultation is to gain preliminary understanding of the ability of Chinese young-old adults to understand and undertake the intervention; and to solicit their feedback on the content and duration of the initial SHEEP health education. Principal participants in the second round of PPI consultation will include a gerontologist, a geriatric nurse, a rehabilitation physician, and a rehabilitation nurse from Xiangya Nursing School and Xiangya Second Affiliated Hospital, Central South University, China. The gerontologist and geriatric nurse should have rich experience of geriatric health education, and their consultation is intended to make the SHEEP health education strategy more systematic, detailed, and in line with lay language requirements for Chinese older adults. Likewise, the purpose of the consultation with the rehabilitation physician and nurse with deep knowledge of geriatric sports medicine is to revise the SHEEP exercise intervention to ensure its safety and suitability for older people. 

d.Please clearly define which criteria you are going to use to define possible sarcopenia

Because this study will be carried out in China, in the section of research participants, we have made it clear that the diagnostic criteria used in this study is the criteria for possible sarcopenia proposed by the Asian Working Group for Sarcopenia (AWGS, 2019). (P8)

e.Addressing possible sarcopenia is a multipronged approach. One intervention - that too unsupervised will not make much of a difference.

Our study is to co-design a multicomponent intervention (health education plus exercise) strategy for preventing possible sarcopenia through behaviour change in community-dwelling young-older adults. The purpose of health education is to improve the target population's understanding of the disease of sarcopenia and teach them ways to prevent sarcopenia, while the purpose of exercise is to help the targeted population to develop the habit of regular exercise to achieve long-term prevention of sarcopenia. 

It is true that we cannot monitor the whole health education and exercise process of participants, which is the shortcoming of using social media for intervention. We will only conduct weekly follow-up by phone or text message, but in fact, this is one of the feasibility we will explore in this research program, that is, how feasible is using social media to carry out health education and exercise intervention? We will conduct a feasibility study in the next stage. The results of this study program can provide valuable reference for the subsequent research.

f. What are the end points - who will be the judge

As this is a study for intervention development, no one will be judge at the end points. We will collect different opinions from different stakeholders about how to modify the intervention strategy, so the qualitative data from PPI consultation, focus groups and semi-structured interview will be the main results, which will further use to refine the intervention strategy. We (the study team) will use these findings to inform a future definitive study.

---

## [Decision Letter · Decision Letter 2]

2 Jan 2024

PONE-D-23-14790R2Social-media based Health Education plus Exercise Programme (SHEEP) to improve muscle function among young-old adults with possible sarcopenia in the community a study protocol for intervention developmentPLOS ONE

Dear Dr. SHI,

Thank you for submitting your manuscript to PLOS ONE. After careful consideration, we feel that it has merit but does not fully meet PLOS ONE’s publication criteria as it currently stands. Therefore, we invite you to submit a revised version of the manuscript that addresses the points raised during the review process.

We have completed the review of your manuscript and a summary is appended below. The reviewer(s) have recommended some minor revisions to your manuscript.  Therefore, I invite you to respond to the reviewer(s)' comments and revise your manuscript.

We look forward to receiving your revised manuscript.

Kind regards,

Sayani Das, PhD

Academic Editor

PLOS ONE

Journal Requirements:

Reviewers' comments:

Reviewer's Responses to Questions

**Comments to the Author**

1. Does the manuscript provide a valid rationale for the proposed study, with clearly identified and justified research questions?

Reviewer #4: Yes

Reviewer #5: Partly

2. Is the protocol technically sound and planned in a manner that will lead to a meaningful outcome and allow testing the stated hypotheses?

Reviewer #4: Yes

Reviewer #5: Yes

3. Is the methodology feasible and described in sufficient detail to allow the work to be replicable?

Reviewer #4: Yes

Reviewer #5: Yes

4. Have the authors described where all data underlying the findings will be made available when the study is complete?

Reviewer #4: Yes

Reviewer #5: No

5. Is the manuscript presented in an intelligible fashion and written in standard English?

Reviewer #4: Yes

Reviewer #5: Yes

6. Review Comments to the Author

You may also provide optional suggestions and comments to authors that they might find helpful in planning their study.

Reviewer #4: The protocol has been well-written with sufficient and detailed description of the methodology. I look forward to the results of the study.

Reviewer #5: Comments:

I have read with interest the manuscript “Social-media based Health Education plus Exercise Programme (SHEEP) to improve muscle function among young-old adults with possible sarcopenia in the community a study protocol for intervention development”.

The manuscript is well-written, and the findings are of utmost importance.

Some minor comments and suggestions can be found below:

• Title page (page 9):

It may be worth to provide the name of the country where the intervention will be conducted.

• Abstract (page 11):

An option may be to define possible sarcopenia in the abstract.

• Keywords (page 11):

Keywords after the abstract are missing.

• Introduction; first paragraph (page 12):

Are there any data on the effects of possible sarcopenia on health and the economic burden of possible sarcopenia?

• Introduction; second paragraph (page 12):

How do the authors define young older adults?

• Introduction; third paragraph (page 12):

A formal definition of health education may be needed.

• Introduction; fourth paragraph (page 13):

Given that TikTok will be used for the intervention of interest, it is critical to provide more information on how TikTok has reached Chinese seniors in prior research.

• Introduction; sixth paragraph (page 14):

Has co-design already been used in the field of sarcopenia?

• Material and methods; Research participants (page 16):

The fact that the intervention will include young older adults already using TikTok may limit the generalizability of the findings of the study. Why have the authors not decided to provide some training to individuals not using TikTok prior to the intervention?

• Material and methods; Sampling method and sample size (page 16):

It is unclear to me why the authors decided to include 45 young older adults specifically.

• Material and methods; Research ethics (page 17):

It is critical to name the Chinese organizations of interest which granted permission for the study.

• Material and methods; Data collection and analysis (page 22):

How will the storage of the data be handled? Besides, in which country will the data be analyzed (e.g., China or the United Kingdom)? Finally, where all data underlying the findings will be made available when the study is complete?

• Discussion; first paragraph (page 23):

It could be interesting for the reader to provide the sample sizes of the different studies mentioned in this paragraph of the discussion.

• Discussion; fourth paragraph (page 24):

It may be worth considering providing more details on how the present intervention will shape future public health interventions and act as a platform for future research.

7. PLOS authors have the option to publish the peer review history of their article (what does this mean?). If published, this will include your full peer review and any attached files.

Reviewer #4: **Yes: **Dr. Sunny Singhal

Reviewer #5: **Yes: **Louis JACOB

---

## [Author Response · Author response to Decision Letter 2]

19 Jan 2024

Response to comments

Dear Editor and Reviewers: 

Manuscript ID number: PONE-D-23-14790R2

Title: Social-media based Health Education plus Exercise Programme (SHEEP) to improve muscle function among young-old adults with possible sarcopenia in the community: a study protocol for intervention development

Thank you so much for your insightful feedback on our manuscript. We have revised and responded to each comment in accordance with your suggestion, as demonstrated below.

Journal Requirements 

1.Please review your reference list to ensure that it is complete and correct. If you have cited papers that have been retracted, please include the rationale for doing so in the manuscript text, or remove these references and replace them with relevant current references. Any changes to the reference list should be mentioned in the rebuttal letter that accompanies your revised manuscript. If you need to cite a retracted article, indicate the article’s retracted status in the References list and also include a citation and full reference for the retraction notice. 

We have reviewed and checked the reference list to ensure that it is complete and correct. 

(P17-21)

Reviewers' Comments

1. Does the manuscript provide a valid rationale for the proposed study, with clearly identified and justified research questions?

Reviewer #4: Yes

Reviewer #5: Partly 

We have added more evidence in the background section according to the suggestion from reviewer 5.

2. Is the protocol technically sound and planned in a manner that will lead to a meaningful outcome and allow testing the stated hypotheses?

Reviewer #4: Yes

Reviewer #5: Yes

3.Is the methodology feasible and described in sufficient detail to allow the work to be replicable?

Reviewer #2: Yes

Reviewer #3: Yes

4.Have the authors described where all data underlying the findings will be made available when the study is complete?

Reviewer #4: Yes

Reviewer #5: No

All of these findings will serve as evidence for our future feasibility study and will be disseminated via publication in peer-reviewed journals and presentation at relevant conferences, and all relevant data will be made available within the manuscript and its Supporting Information files, in order to serve as valuable resources, references, and directions for other researchers in related fields. (P17)

5.Is the manuscript presented in an intelligible fashion and written in standard English?

Reviewer #4: Yes

Reviewer #5: Yes

6.Review Comments to the Author

Reviewer #4: The protocol has been well-written with sufficient and detailed description of the methodology. I look forward to the results of the study.

Thank you so much for your encouragement.

Reviewer #5: Comments:

I have read with interest the manuscript. The manuscript is well-written, and the findings are of utmost importance. Some minor comments and suggestions can be found below:

Thank you so much for your valuable advice and patient guidance. We have responded to your comments carefully.

• Title page (page 9):

It may be worth to provide the name of the country where the intervention will be conducted.

We have clarified in the title that the study was conducted in China. 

“Social-media based Health Education plus Exercise Programme (SHEEP) to improve muscle function among community-dwelling young-old adults with possible sarcopenia in China: a study protocol for intervention development” (P1,3)

• Abstract (page 11):

An option may be to define possible sarcopenia in the abstract.

Thank you for your advice. We have already described the definition of possible sarcopenia in the background section. We have also added the definition of possible sarcopenia in the abstract according to your suggestion.

Possible sarcopenia refers to low muscle strength. (P3)

• Keywords (page 11):

Keywords after the abstract are missing.

Thank you for your advice. We have added key words in the abstract.

Key words: possible sarcopenia; older adults; community; social media; TikTok; health education; exercise; co-design; intervention development (P3)

• Introduction; first paragraph (page 12):

Are there any data on the effects of possible sarcopenia on health and the economic burden of possible sarcopenia?

Numerous evidence have shown that confirmed and severe sarcopenia is associated with a variety of adverse outcomes in older adults, such as increased risks of falls and fractures, functional decline, disability, lower quality of life, and higher risks of mortality. Untreated sarcopenia may result in significant personal, social, and economic burdens, which has been clarified in our previous scoping review [10]. Nevertheless, there is limited evidence regarding the impact of possible sarcopenia on health and the associated economic cost, warranting further investigation.

• Introduction; second paragraph (page 12):

How do the authors define young older adults?

We have already clarified how we defined “young-old adults” in the section of “research participants”.

Although different scholars interpreted the definition of "young-old" differently, such as 60-69 years or 65-74 years [51,52], based on the fact that this study will be conducted in China, 60-69 years is considered young-old for the purposes of this study. (P8)

In fact, in the original version, we explained in detail why we chose 60-69 as young-old adults in this study. Due to the word limit, we simplified the description, and the detailed description is as follows.

Although age definition of an older person is not uniform to some extent around the world according to different conditions by different countries (e.g.≥50 in Africa, ≥60 in United Nations and China, ≥65 in western countries, ≥75 in Japan), the ages of 60 and 65 years are often used [A1-A3]. In addition, different scholars also have different interpretations of the definition of young-old, like 60-69 or 65-74 [A4-A5]. Based on the study shortage of 60-69 age group from my scoping review and also considering this feasible study will be carried out in China, 60-69 years old is defined as young-old in this study.

[A1] Kowal P, Dowd JE. Definition of an older person. Proposed working definition of an older person in Africa for the MDS Project. Geneva: World Health Organization 2001.

[A2] Orimo H, Ito H, Suzuki T, et al. Reviewing the definition of “elderly”. Geriatrics & Gerontology International 2006; 6(3): 149-158.

[A3] Ouchi Y, Rakugi H, Arai H, et al. Redefining the elderly as aged 75 years and older: Proposal from the Joint Committee of Japan Gerontological Society and the Japan Geriatrics Society. Geriatr Gerontol Int 2017; 17: 1045-1047.

[A4] Forman DE, Berman AD, McCabe CH, et al. PTCA in the elderly: The "young-old" versus the "old-old". Journal of the American Geriatrics Society 1992; 40 (1): 19-22.

[A5] Zizza CA, Ellison KJ, Wernette CM. Total Water Intakes of Community-Living Middle-Old and Oldest-Old Adults. The Journals of Gerontology Series A: Biological Sciences and Medical Sciences 2009; 64A (4): 481-486. 

• Introduction; third paragraph (page 12):

A formal definition of health education may be needed.

Thank you for your advice. We have added the definition of health education.

Health Education, as defined by the World Health Organization, refers to the deliberate creation of learning opportunities that involve communication aimed at enhancing health literacy. This includes improving knowledge and developing life skills that promote individual and community health [11]. (P4)

• Introduction; fourth paragraph (page 13):

Given that TikTok will be used for the intervention of interest, it is critical to provide more information on how TikTok has reached Chinese seniors in prior research.

We have added more references about how TikTok has reached older adults on health information.

A qualitative research showed that during the COVID-19 pandemic, WeChat and TikTok played a significant role in providing older Chinese adults with access to valuable health information; and many older adults were able to modify their health behaviours after incorporating this information and knowledge into their daily lives [27]. Kassamali et al. [28] reported that TikTok held great potential as a platform for disseminating educational information about disease, as their study indicated that the number of views for the hashtag such as #acne, #alopecia, #cyst, #rosacea, and #psoriasis on TikTok doubled in just 5 months. Nonetheless, there are currently no studies utilizing social media for sarcopenia prevention. (P5-6)

• Introduction; sixth paragraph (page 14):

Has co-design already been used in the field of sarcopenia?

We have added a sentence in the sixth paragraph as following:

“There is currently no evidence of co-design in the field of sarcopenia research, making the exploration of this study particularly valuable.” (P6-7)

• Material and methods; Research participants (page 16):

The fact that the intervention will include young older adults already using TikTok may limit the generalizability of the findings of the study. Why have the authors not decided to provide some training to individuals not using TikTok prior to the intervention?

Insufficient time during the intervention development phase prevented us from adequately training a cohort of older individuals who were unfamiliar with TiKTok, which may be a limitation of this study. We also mentioned this in the limitations section of this study. However, our forthcoming feasibility study will offer training to older persons who are eager to utilize TikTok before the intervention, in order to expand the range of recruitment.

Older adults who are unfamiliar with TikTok or lack Internet and modern communication technology access or proficiency were excluded from the recruitment stage, which may result in a less diverse sample. (P16)

• Material and methods; Sampling method and sample size (page 16):

It is unclear to me why the authors decided to include 45 young older adults specifically.

We have clarified clearly in the section of “sample size”. 45 is the maximum number of recruits and is made up of two parts. The sample size was calculated with reference to similar studies about focus group and co-design research.

We aim to recruit a maximum sample of 45 older adults in total. For developing phase, 18~25 older adults will be recruited to participate in focus groups (with a 10~15% attrition rate), and then invited to take part in both the first and the second round of focus groups. Each round will consist of three focus groups, with 18~25 participants divided into three groups of at least six members each, which is based on the recommendations of Leask et al.[50] that 10~12 co-creators in total is advised in co-design research and the Guidelines for Conducting a Focus Group that suggests six to ten people will be appropriate for a focus group [55]. Besides, we require 15~20 additional participants for the pre-test phase. In the two phases of our study, participants have the option of participating in only focus groups, only pre-test, or both focus groups and pre-test. (P8-9)

• Material and methods; Research ethics (page 17):

It is critical to name the Chinese organizations of interest which granted permission for the study.

We have added the name of the Chinese organization.

This study has been approved by the University of Manchester Research Ethics Committee (Project ID: 15664), and permissions have already been granted by collaborators in the Community Nursing Department of Xiang Ya Nursing School, Central South University, China. (P10)

• Material and methods; Data collection and analysis (page 22):

How will the storage of the data be handled? Besides, in which country will the data be analyzed (e.g., China or the United Kingdom)? Finally, where all data underlying the findings will be made available when the study is complete?

We have clarified this part as following:

All data will be saved on servers or computers that are certified by the University of Manchester and have safe backup systems. The data will be analyzed in China, in accordance with the data management plan. (P14)

All of these findings will serve as evidence for our future feasibility study and will be disseminated via publication in peer-reviewed journals and presentation at relevant conferences, and all relevant data will be made available within the manuscript and its Supporting Information files, in order to serve as valuable resources, references, and directions for other researchers in related fields. (P17)

• Discussion; first paragraph (page 23):

It could be interesting for the reader to provide the sample sizes of the different studies mentioned in this paragraph of the discussion.

We have added the sample sizes of the different studies mentioned in this paragraph of the discussion.

One study in China involved a 12-week intervention of 234 older adults with sarcopenia and revealed that nutrition interventions through an app were effective in optimising the diets and increasing the skeletal muscle mass [63]. Another study conducted a randomized controlled trial on 23 participants and indicated that supervised resistance exercise through a video conferencing software (Skype™) had positive effects on sarcopenia-related factors among community-dwelling senior citizens in South Korea [64]. (P16)

• Discussion; fourth paragraph (page 24):

It may be worth considering providing more details on how the present intervention will shape future public health interventions and act as a platform for future research.

We have added more details according to your suggestion as following:

Furthermore, the comprehensive co-design procedure will build a foundation for co-design research in the field of sarcopenia. Additionally, the health education and exercise video resources on TikTok will serve as valuable reference materials for future intervention by other researchers. (P17)

Finally, we really appreciate all the time and effort you put into this document, as well as all your insightful suggestions for revising the manuscript to make it more readable, clear, and academic. Thank you so much once more!

Best wishes

---

## [Decision Letter · Decision Letter 3]

7 Feb 2024

Social-media based Health Education plus Exercise Programme (SHEEP) to improve muscle function among community-dwelling young-old adults with possible sarcopenia in China: a study protocol for intervention development

PONE-D-23-14790R3

Dear Dr. SHI,

We’re pleased to inform you that your manuscript has been judged scientifically suitable for publication and will be formally accepted for publication once it meets all outstanding technical requirements.

Kind regards,

Sayani Das, PhD

Academic Editor

PLOS ONE

Additional Editor Comments (optional):

Reviewers' comments:

Reviewer's Responses to Questions

**Comments to the Author**

1. Does the manuscript provide a valid rationale for the proposed study, with clearly identified and justified research questions?

Reviewer #4: Yes

Reviewer #5: Yes

2. Is the protocol technically sound and planned in a manner that will lead to a meaningful outcome and allow testing the stated hypotheses?

Reviewer #4: Yes

Reviewer #5: Yes

3. Is the methodology feasible and described in sufficient detail to allow the work to be replicable?

Reviewer #4: Yes

Reviewer #5: Yes

4. Have the authors described where all data underlying the findings will be made available when the study is complete?

Reviewer #4: Yes

Reviewer #5: Yes

5. Is the manuscript presented in an intelligible fashion and written in standard English?

Reviewer #4: Yes

Reviewer #5: Yes

6. Review Comments to the Author

You may also provide optional suggestions and comments to authors that they might find helpful in planning their study.

Reviewer #4: The protocol has been well-written with sufficient and detailed description of the methodology. I look forward to the results of the study.

Reviewer #5: Comments to the authors:

I have read with interest the revised version of the manuscript “Social-media based Health Education plus Exercise Programme (SHEEP) to improve muscle function among young-old adults with possible sarcopenia in the community a study protocol for intervention development”.

The authors have answered all my comments, and I thank them for their critical work.

My recommendation is to accept this valuable manuscript.

7. PLOS authors have the option to publish the peer review history of their article (what does this mean?). If published, this will include your full peer review and any attached files.

Reviewer #4: **Yes: **Dr. Sunny Singhal

Reviewer #5: **Yes: **Louis Jacob

---

## [Editor Report · Acceptance letter]

20 Mar 2024

PONE-D-23-14790R3 

PLOS ONE

Dear Dr. SHI, 

I'm pleased to inform you that your manuscript has been deemed suitable for publication in PLOS ONE. Congratulations! Your manuscript is now being handed over to our production team.

Kind regards, 

on behalf of

Dr Sayani Das 

Academic Editor

PLOS ONE